# MiniGPT-3D: Efficiently Aligning 3D Point Clouds with Large Language Models using 2D Priors

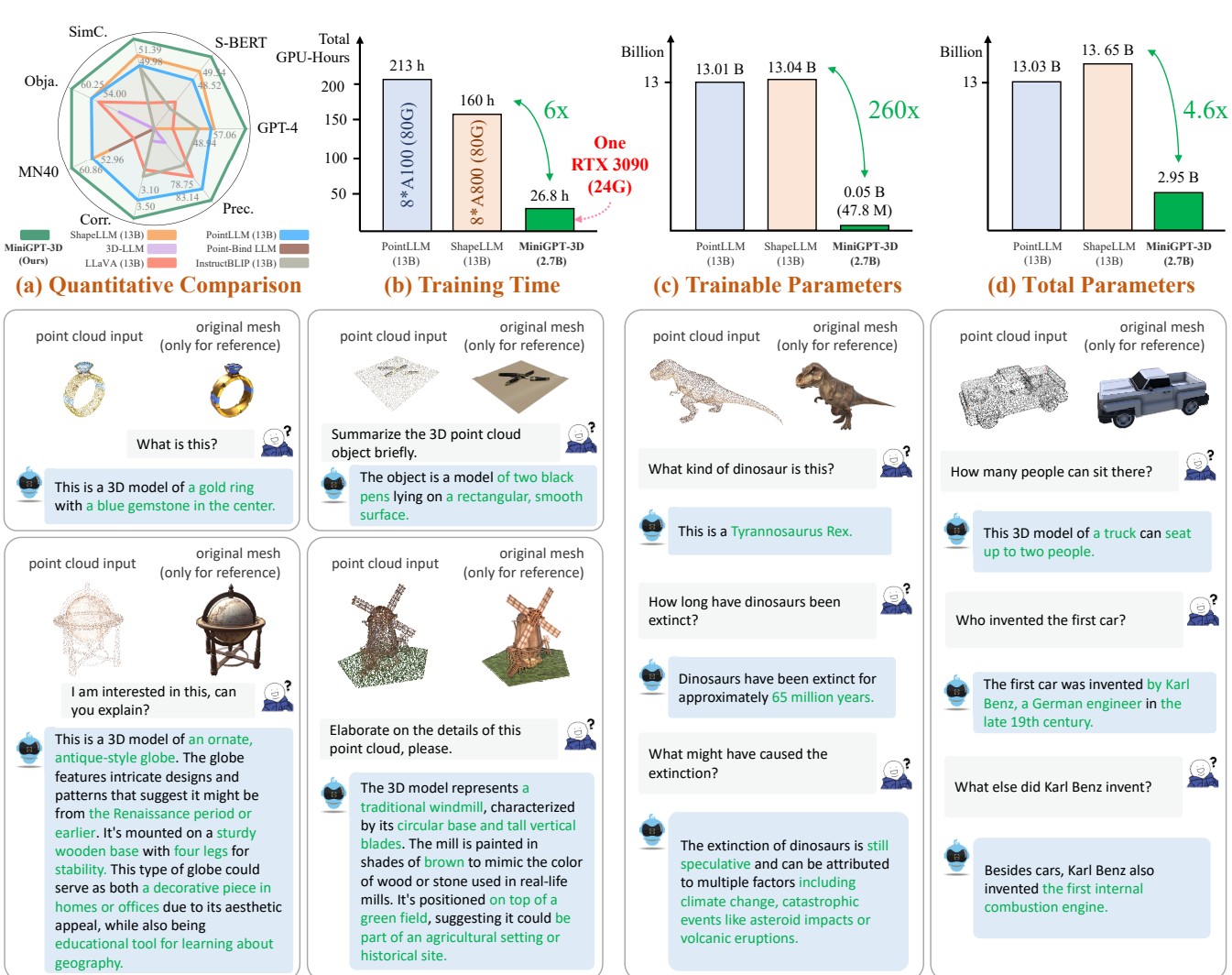

Figure 1: Demonstrations of MiniGPT-3D. We present MiniGPT-3D, an efficient and powerful 3D-LLM that aligns 3D point clouds with large language models using 2D priors from large 2D vision-language models. This figure demonstrates MiniGPT-3D's superior performance and efficient training compared to existing 3D-LLMs. We also show some prediction examples in 3D recognition, captioning, and question-answering tasks, with the correct and fine-grained answers highlighted in green.

**Unpublished working draft. Not for distribution.**

## ABSTRACT

Large 2D vision-language models (2D-LLMs) have gained significant attention by bridging Large Language Models (LLMs) with images using a simple projector. Inspired by their success, large 3D point cloud-language models (3D-LLMs) also integrate point

clouds into LLMs. However, directly aligning point clouds with LLM requires expensive training costs, typically in hundreds of GPU-hours on A100, which hinders the development of 3D-LLMs. In this paper, we introduce **MiniGPT-3D**, an efficient and powerful 3D-LLM that achieves multiple **SOTA** results while training for only **27 hours on one RTX 3090**. Specifically, we propose to align 3D point clouds with LLMs using 2D priors from 2D-LLMs, which can leverage the similarity between 2D and 3D visual information. We introduce a novel four-stage training strategy for modality alignment in a cascaded way, and a mixture of query experts module to adaptively aggregate features with high efficiency. Moreover, we utilize parameter-efficient fine-tuning methods LoRA and Norm fine-tuning, resulting in only **47.8M** learnable parameters, which is up to 260× fewer than existing methods. Extensive experiments show that MiniGPT-3D achieves SOTA on 3D object classification and captioning tasks, with significantly cheaper training costs. Notably, MiniGPT-3D gains an 8.12 increase on GPT-4 evaluation score for the challenging object captioning task compared to ShapeLLM-13B, while the latter costs 160 total GPU-hours on 8 A800. We are the first to explore the efficient 3D-LLM, offering new insights to the community. We will release the code and weights after review.

## CCS CONCEPTS

• **Computing methodologies** → **Computer vision**; **Natural language processing**.

## KEYWORDS

Multimodal Large Language Models, Efficiently Multimedia Alignment, 3D Point Cloud Understanding

## 1 INTRODUCTION

Large Language Models (LLMs) have recently driven advancements in multiple fields [15, 35, 45, 46], benefiting from their world knowledge. Built on LLMs, large 2D vision-language models (2D-LLMs) [4, 27, 62] can align image features with text through an image feature projector, enabling 2D-LLMs to understand visual content. Inspired by 2D-LLMs, large 3D point cloud-language models (3D-LLMs) [39, 40, 51] aim to incorporate 3D point cloud features into LLMs, equipping LLMs with the ability to perceive and reason in 3D space. These 3D-LLMs hold promise for widespread applications in fields like robotics [44, 48] and autonomous driving [10, 15]. However, 3D-LLMs are expensive to train. For example, training PointLLM-13B [51] takes 213 total GPU-hours on 8 A100 GPU, making research and applications extremely challenging. Here, we aim to find a more efficient way to connect 3D point clouds with LLMs.

We observe that existing 3D-LLMs directly align point cloud encoders with LLMs. Although these encoders can produce somewhat unified features through multimodal pre-training, there is still a significant modality gap between 3D points with LLMs, requiring substantial resources for alignment. Besides, in contrast to resource-intensive alignment between vision and language, 3D point clouds and 2D images are both visual modalities, which makes it easier to align their representations. Thus, we pose a question: ***Can we use 2D-LLMs as a strong prior to connect LLMs and 3D data, making alignment more efficient?*** In other words, as shown in Figure 2, leveraging pre-trained 2D-LLMs directly allows for

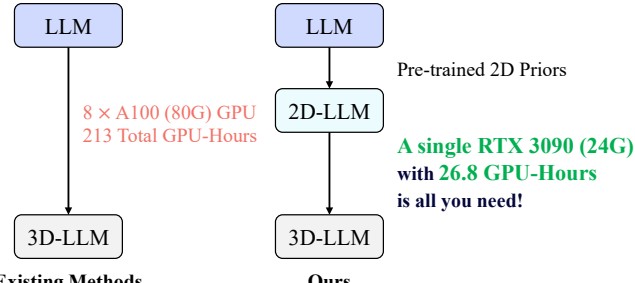

**Figure 2: Existing methods and ours to align 3D with LLMs.**

cutting down the cost of vision-language alignment, leaving only the 2D-3D vision alignment, which is significantly cheaper.

Following this intuition, we propose MiniGPT-3D, an efficient 3D-LLM that connects 3D point clouds and LLMs using 2D-LLMs as priors. Our MiniGPT-3D achieves multiple state-of-the-art (SOTA) results, requiring only 27 hours of training on a single RTX 3090 GPU. Specifically, we propose an efficient four-stage training strategy in a cascaded way, gradually allowing the model to learn unified visual-textual representations. This process achieves the smooth transfer of priors from 2D-LLM to the 3D space, thus efficiently constructing a bridge from 3D to LLM. Moreover, we introduce the Mixture of Query Experts (MQE), which comprises multiple query experts and an expert router, enabling the adaptive aggregation of features from multiple experts with only 0.4M parameters. MQE dynamically adjusts the cooperation relationship between experts, thereby aggregating 3D features from multiple perspectives into the semantic space of 2D-LLM. Meanwhile, we employ various parameter-efficient fine-tuning (PEFT) technologies like LoRA [21] and Norm fine-tuning, and utilize an efficient LLM, further reducing the model's training overhead.

As shown in Figure 1, MiniGPT-3D achieves new SOTA performance on generative 3D object classification and object captioning tasks. Specifically, compared to the powerful baseline ShapeLLM-13B [39], MiniGPT-3D achieves a 6.77% increase in classification average accuracy and an 8.12 increase in GPT-4 evaluation score. Notably, MiniGPT-3D utilizes extremely cheaper training resources (1× RTX 3090 vs. 8× A800), with up to 6× acceleration (26.8h on RTX 3090 vs. 160h on A800). Furthermore, our model has significantly fewer trainable parameters, reduced by up to 260×, with 2.95B model parameters in total, which is decreased by up to 4.6×.

MiniGPT-3D takes the first step in efficient 3D-LLM, we hope that MiniGPT-3D can bring new insights to this community. In summary, our contributions are as follows:

- We present MiniGPT-3D, an efficient and powerful 3D-LLM that aligns 3D points with LLMs using 2D priors, achieving multiple SOTA with only 26.8h of training on one RTX 3090.
- We propose an efficient four-stage training strategy in a cascaded way, gradually transferring the knowledge from 2D-LLMs to 3D while requiring only 47.8M learnable parameters.
- We design the mixture of query experts to aggregate multiple features from different experts with only 0.4M parameters.
- Extensive experiments show the superior performance of MiniGPT-3D on multiple tasks while reducing the training time and parameters by up to 6x and 260x, respectively.

**Figure 3: Training framework and strategy. Our MiniGPT-3D utilizes a four-stage training strategy. (a) We solely train the point cloud projection layer (MLP). (b) We train the modality projector while fine-tuning the point cloud projection layer, Q-Former, and LLM backbone. (c) We further enhance the modules trained in the second stage by leveraging a more challenging task. (d) Finally, we only train the mixture of query experts, while freezing the remaining modules.**

## 2 RELATED WORK

### 2.1 Large 2D Vision-Language Models

The exceptional instruction-following and generalization capabilities of LLMs [46, 49, 53, 55] have been integrated into vision, leading to the emergence of large 2D vision-language models (2D-LLMs). Early works such as Flamingo [1] and BLIP-2 [27] successfully use projectors to align vision information to LLMs. More recently, most works mainly focus on improving model capabilities through expanding the instruction-tuning dataset [5, 30, 61], increasing resolution of image [2, 31], enhancing image encoders [7, 59]. Meanwhile, some methods [8, 9, 57, 60] have also begun to explore efficient 2D-LLM. Models like TinyLlama [60] and TinyGPT-V [57] use Phi-2 [33], an efficient LLM, to achieve easily deployable 2D-LLMs. Among them, TinyGPT-V leverages LoRA [21] technology and pre-trained modules to achieve extremely efficient fine-tuning. However, TinyGPT-V can only handle 2D images, efficient 3D-LLM remains unexplored, and we aim to fill this gap.

### 2.2 Large 3D Point Cloud-Language Models

Large 3D point cloud-language models (3D-LLMs) introduce LLM into the point cloud modality [6, 20, 23, 29, 36, 39, 40, 51, 54]. Early attempt [20] renders 3D objects into 2D multi-view images, then utilizes 2D-LLM to understand 3D. However, the absence of direct perception of raw point cloud data limits its comprehension of 3D geometry. To address this issue, recent works [6, 23, 36, 40] propose to discard the "rendering" and encode point cloud directly, followed by modal alignment to fixed LLMs via trainable projectors. PointLLM [51] and ShapeLLM [39] show that models can be enhanced after fully fine-tuning. However, the training of 3D-LLMs

is expensive. For instance, PointLLM-13B requires training on 8 A100 GPUs for up to 216 total GPU-hours. We observe that with 2D-LLM as visual prior, we can not only bypass the "point cloud rendering", but also make this hierarchical alignment extremely efficient. Therefore, we propose MiniGPT-3D, different from existing 3D-LLMs which aligns 3D points directly to LLMs, our MiniGPT-3D leverages the powerful priors from 2D-LLM as a linkage between LLM and 3D points, using only a RTX 3090 to train for 27 hours.

### 2.3 Mixture of Experts

Mixture of Experts (MoE) [22, 24] is an ensemble learning technique that adaptively activates selected modules, referred to as experts, based on input. MoE is widely used in various fields [14, 25, 26, 42, 43]. Shazeer et al. introduce MoE into NLP for the first time, where each intervening layer between LSTM layers serves as an expert. Gshard [26] further expands the MoE to Transformer [47], treating each Feed-Forward Neural Network (FNN) as an expert. Recently, with the emergence of LoRA, several works [13, 16, 58] design FFN's LoRA network as an expert to efficiently fine-tune LLM. Moreover, OneLLM [19] introduces MoE to the learned projector of 2D-LLM, with each projector serving as an expert. In our work, we integrate the MoE concept into the queries of Q-Former [27], treating each set of queries as an expert. These experts adaptively aggregate point cloud features across diverse extraction perspectives.

## 3 METHOD

In this section, we first introduce the architecture of MiniGPT-3D (Sec. 3.1), and then present our four-stage training strategy (Sec. 3.2), and finally elucidate the training loss for MiniGPT-3D (Sec. 3.3).

## 3.1 Model Architecture

Figure 3 depicts the architecture of MiniGPT-3D, which consists of the six main components: a point cloud encoder, a point cloud projection layer (MLP), a Q-Former, a mixture of query expert (MQE), a modality projector, and a large language model.

The MiniGPT-3D framework introduces a two-step projection process, transforming the point cloud from 3D to 2D and then to 1D. Specifically, the point cloud is passed to the point cloud encoder to extract 3D features. Subsequently, features are then projected into a 2D semantic space using the point cloud projection layer. Finally, leveraging the 2D-LLM modules including the Q-Former, modality projector, Norm of LLM, and LoRA of LLM, features in 2D-LLM space are transduced into the 1D-text space of LLM, enabling efficient alignment between 3D and LLM. Additionally, MQE enhances MiniGPT-3D's comprehensive and accurate perception of 3D objects. Details are presented in the following sections.

*3.1.1* **3D Features to 2D**. During this process, the point cloud is encoded into 3D features and subsequently projected into the 2D semantic space of the 2D-LLM.

**Point Cloud Encoder**. The input point cloud is encoded into 3D features by the point cloud encoder $f_{pc}$. Specifically, the point cloud $P \in \mathbb{R}^{n \times d}$ is input to $f_{pc}$, where $n$ is the number of points and $d$ denotes the feature dimension of each point. Then, $f_{pc}$ outputs a point feature sequence $X \in \mathbb{R}^{m \times b}$, comprising $m$ features, each with a dimension of $b$. In our experiments, we employ the Point-BERT [56] model, pre-trained on ULIP-2 [52] using the Objaverse [12] dataset, as the point cloud encoder. To maintain pre-training knowledge, we freeze the encoder's parameters on all training stages.

**Point Cloud Projection Layer**. The point cloud projection layer $f_{MLP}$ is an MLP with two linear layers, which embeds point features $X$ into the semantic space of the pre-trained 2D Q-Former [27], aligning their dimensions. Concisely, $Y = f_{MLP}(X)$, where $Y \in \mathbb{R}^{m \times b'}$ and $b'$ is the hidden space dimension of Q-Former.

*3.1.2* **Features in 2D-LLM space to LLM**. This part transduces the point cloud representation in the 2D semantic space of 2D-LLM to the 1D text space of LLM.

**Q-Former**. The Q-Former $f_{QF}$, with a decoder-based Transformer structure, transforms point features $Y$ into point queries $\overline{Q}$. This process not only enhances the information extracted from point cloud features but also reduces input size for subsequent LLM, accelerating training and inference. Concisely, $\overline{Q} = f_{QF}(Y, Q)$, where $Q \in \mathbb{R}^{o \times b'}$, $\overline{Q} \in \mathbb{R}^{o \times b'}$. $Q$ is the queries of Q-former and $o$ is the number of query. In experiments, we initialize Q-Former with BLIP-2 [27] pre-trained weights. Given Q-Former's extensive 105M parameters, we employ PEFT technologies to fine-tune its Query, Key, and Value layers, and normalization layers, thus enhancing adaptability to point clouds while preserving 2D knowledge.

**Mixture of Query Experts**. Inspired by multi-view image rendering for 3D-to-2D projection, we propose the Mixture of Query Experts (MQE) to achieve a similar effect. In the process of MQE, multiple sets of queries (query expert) are used to transform point features into the semantic space of 2D Q-Former. MQE is the first

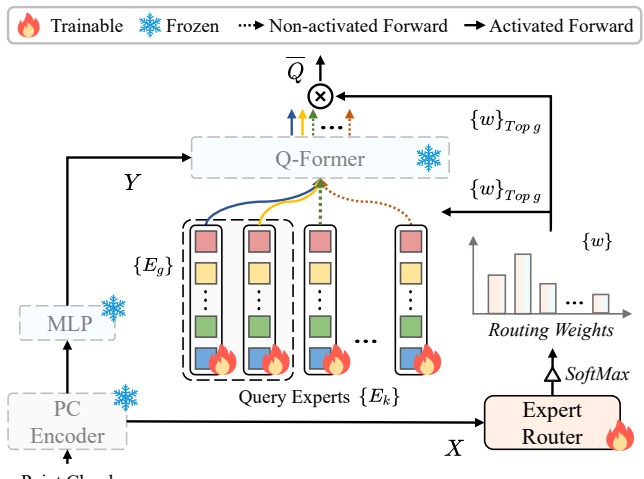

**Figure 4: The framework of the mixture of query experts. First, a point cloud is encoded to features $X$ and $Y$. Feature $X$ is then passed through to the expert router, assigning softmax-based weights to experts. The top $g$ experts are selected based on these weights. These experts, together with $Y$, are then fed into the Q-Former, and their outputs are weighted to produce the final point queries $\overline{Q}$.**

to introduce dynamic routing of MoE into queries, enabling adaptive activation of more suitable query experts to capture richer semantic information across diverse point cloud inputs, as shown in Figure 4. MQE contains $k$ trainable query experts $\{E_k\}$, each is a set of queries initialized from BLIP-2. To integrate multiple query experts into one set of queries, we use a dynamic routing, expert router $f_R$, which regulates each expert's contribution. The expert router is an MLP that accepts feature $X$ and assigns routing weights to each expert. We employ the sparse routing strategy [43], selecting $g$ experts with the highest weights. Subsequently, the selected query experts $\{E_g\}$ utilize Q-Former to extract high-dimensional semantics $\{\overline{Q_h}\}$ from the feature $Y$. $\{\overline{Q_h}\}$ are then weighted by the corresponding routing weights to generate the final point queries $\overline{Q}$. The process can be formulated as:

$$\overline{Q} = \sum_{E_q \in \{E_g\}} w_q \cdot f_{QF}(Y, E_q), \quad (1)$$

$$w_q = \text{Softmax}\left(f_R(X)\right)[q]. \quad (2)$$

To enable query experts to learn knowledge within a stable 3D-LLM semantic context, MQE is only utilized in the final training stage, by which time other modules have completed training.

**Modality Projector**. We use an MLP as the modality projector to bridge the modality gap between point cloud and text, while transforming point queries $\overline{Q} \in \mathbb{R}^{o \times b'}$ into point tokens $T_{pc} \in \mathbb{R}^{o \times c}$, where $c$ denotes the shared dimension of both point and text tokens.

*3.1.3* **Large Lanuguage Model Backbone**. To minimize GPU memory usage during training, we utilize Phi-2 [33] with 2.7 billion parameters as the large language model backbone of MiniGPT-3D. In MiniGPT-3D, the LLM backbone $f_{llm}$ processes a sequence of tokens $T = (t_1, t_2, \ldots, t_j) \in \mathbb{R}^{j \times c}$, where $j$ is the number of

**Table 1: Each training stage setups and overhead.**

| Training Stages | Dataset Types | Dataset Scale | Epochs | Init_lr & Min_lr | Trainable Parameters | Training Time using One RTX 3090 GPU |
|---|---|---|---|---|---|---|
| Stage I | Brief Caption | 660 k | 1 | 3e-5, 1e-5 | 1.4 M | 9.4 h |
| Stage II | Brief Caption | 660 k | 1 | 3e-5, 1e-5 | 47.4 M | 10.9 h |
| Stage III | Detailed Caption & Conversation | 70 k | 3 | 1e-5, 1e-6 | 47.4 M | 4.9 h |
| Stage IV | Detailed Caption & Conversation | 70 k | 1 | 5e-6, 1e-6 | 0.4 M | 1.6 h |

tokens, including point tokens and text tokens. Leveraging the self-attention mechanism, the LLM backbone can comprehend the semantic relationships from different modality tokens and generate responses for given instructions. This process can be expressed as:

$$\hat{T} = f_{llm}(T), \qquad (3)$$

where $\hat{T} = (\hat{t}_1, \hat{t}_2, \ldots, \hat{t}_j) \in \mathbb{R}^{j \times c}$, and $\hat{t}_i$ denotes the predicted $i$-th token, based on the semantics of all previous tokens $\{t_{<i}\}$. Subsequently, $\hat{t}_i$ is passed through a linear layer $f_{llm \rightarrow vocab}$ to be mapped into the vocabulary space. A softmax operation is then applied to compute a probability distribution across the vocabulary, with the word of highest probability designated as the prediction $z_i$ for $\hat{t}_i$. The process can be formulated as:

$$\tilde{t}_i = f_{llm \rightarrow vocab}(\hat{t}_i), \qquad (4)$$

$$z_i = arg \max_{w \in vocab} \text{Softmax}(\tilde{t}_i)[w]. \qquad (5)$$

As LLMs are primarily trained on text, a perception gap arises when processing non-textual information. Therefore, we adapt PEFT technology LoRA [21] to the LLM backbone, and also further fine-tune the normalization layers, preserving learned knowledge and reducing computational overhead.

## 3.2 Training Stages

To gradually transfer the priors of 2D-LLM to point cloud modality and enhance the nascent 3D-LLM's comprehension, our training process includes four stages, each focusing on a distinct task, as shown in Figure 3. The following subsections will describe them.

*3.2.1* **Stage I**. As shown in Figure 3(a), the first stage aims to bridge the knowledge gap between the 3D point cloud encoder and 2D-LLM modules, facilitating a seamless transition from 3D to 2D. We solely train the point cloud projection layer (MLP), with other modules frozen. Initialization is sourced from ULIP-2 [52] for the encoder, BLIP-2 [27] for Q-Former, and TinyGPT-V [57] for normalization layers of LLM, LoRA of LLM, and the modality projector. Since the frozen Q-Former from BLIP-2 is also used in TinyGPT-V, MiniGPT-3D only owns two knowledge domains from 3D of ULIP-2 and 2D-LLM of TinyGPT-V before training. To build a robust bridge between domains, we train the projection layer using 660k caption-point cloud pairs, involving 1.4M parameters, as detailed in Table 1.

*3.2.2* **Stage II**. In the second stage, our objective is to transfer the vision-language knowledge domain to 3D space, establishing the 3D-language knowledge domain. As shown in Figure 3(b), we fine-tune four parts: the point cloud projection layer (MLP), the Q-Former, the modality projector, and the LLM. Utilizing the 3D-2D bridge of the first stage, 2D-LLM modules, via fine-tuning, gain comprehension of 3D point clouds and gradually transfer the powerful priors to be

the 3D-language knowledge. During this process, to minimize the impact of the 3D-2D bridge, we employ the identical dataset from the first stage to train 47.4M parameters, as outlined in Table 1.

*3.2.3* **Stage III**. To gain better 3D-language knowledge, we further fine-tune the modules trained in the second stage and utilize a more challenging dataset, including detailed caption-point cloud pairs and conversations, to empower MiniGPT-3D with the capabilities to comprehend and respond to complex instructions.

*3.2.4* **Stage IV**. During the prior stages, using a single set of queries restricts 3D perception perspective, leading to incomplete cognition. To refine MiniGPT-3D's perception, we introduce MQE to adaptively activate suitable multiple query experts for Q-Former, as shown in Figure 3(d). Distinct from the preceding three stages focusing on rapidly establishing 3D-language knowledge, this stage presents a stable semantic context for query experts to learn knowledge efficiently. Specifically, we only fine-tune 0.4M MQE-related parameters, reusing the dataset from the third stage to minimize the impact of data distribution changes, as outlined in Table 1.

## 3.3 Training Objective

The training objective of MiniGPT-3D aims to minimize the discrepancy between predicted and true probability distributions at each token position. Given a point cloud and corresponding text instruction, MiniGPT-3D outputs a sequence $\hat{T}$. Next, $\hat{T}$ is processed by $f_{llm \rightarrow vocab}$ and then a softmax operation is applied to obtain the probability distribution over the vocabulary for each output token, denoted as $\overline{T}$. The training loss is formulated as follows:

$$\mathcal{L} = \text{CrossEntropy}\left(h(G), \overline{T}\right), \qquad (6)$$

where the $h(\cdot)$ represents the LLM's tokenizer. $G$ is the ground truth text. The $CrossEntropy(\cdot)$ refers to the cross-entropy loss function. Notably, we only compute the loss for the generated text.

## 4 EXPERIMENTS

### 4.1 Experimental Settings

Utilizing one RTX 3090 GPU with 24GB of RAM, we train MiniGPT-3D with only 47.8M trainable parameters in 26.8 hours. We adopt the AdamW optimizer with a weight decay of 0.05 and a cosine decay with linear warm up learning rate schedule. The initial learning rate decreases gradually as the training stage advances, as shown in Table 1. We use the point-text instruction dataset [51], including 660K brief-description instructions and 70K complex instructions. 200 objects are splited as test data, following PointLLM [51] and ShapeLLM [39]. For each input point cloud $P \in \mathbb{R}^{n \times d}$, the number of point $n$ is 8192, and the dimension $d$ is 6. We default point clouds without color to black. For a fair comparison, we adopt the identical versions models of GPT-4 [35] ("gpt-4-0613") and ChatGPT [34]

**Table 2: Generative 3D object classification results on the ModelNet40 test split and Objaverse. The accuracy (%) under the Instruction-typed (I) prompt "What is this?" and the Completion-type (C) prompt "This is an object of" are reported. The bold and underline indicate the best and second best results, respectively.**

| Model | Reference | LLM Size | Trainable Params | Input | ModelNet40 | | | Objaverse | | | Average |
|---|---|---|---|---|---|---|---|---|---|---|---|
| | | | | | (I) | (C) | Average | (I) | (C) | Average | |
| InstructBLIP-7B [11] | NeurIPS,23 | 7B | 0.20B | Single-V. Img. | 19.53 | 31.48 | 25.51 | 45.00 | 42.00 | 43.50 | 34.50 |
| InstructBLIP-13B [11] | NeurIPS,23 | 13B | 0.20B | Single-V. Img. | 25.97 | 31.40 | 28.69 | 37.00 | 31.50 | 34.25 | 31.47 |
| LLaVA-7B [32] | NeurIPS,23 | 7B | 7.03B | Single-V. Img. | 39.75 | 39.67 | 39.71 | 49.50 | 50.50 | 50.00 | 44.86 |
| LLaVA-13B [32] | NeurIPS,23 | 13B | 13.03B | Single-V. Img. | 37.12 | 36.06 | 36.59 | 53.00 | 50.50 | 51.75 | 44.17 |
| 3D-LLM [20] | NeurIPS,23 | 13B | - | 3D Obj. + Mul.-V. Img. | - | - | - | 49.00 | 41.50 | 45.25 | 45.25 |
| Point-Bind LLM [18] | arXiv,23.9 | 7B | - | 3D Point Cloud | 51.90 | 39.71 | 45.81 | 6.00 | 4.50 | 5.25 | 25.53 |
| PointLLM-7B [51] | arXiv,23.8 | 7B | 7.01B | 3D Point Cloud | 53.44 | 51.82 | 52.63 | 55.00 | 51.00 | 53.00 | 52.82 |
| PointLLM-13B [51] | arXiv,23.8 | 13B | 13.01B | 3D Point Cloud | 53.00 | 52.55 | 52.78 | 56.50 | 51.50 | 54.00 | 53.39 |
| ShapeLLM-7B [39] | arXiv,24.2 | 7B | 7.04B | 3D Point Cloud | - | - | 53.08 | - | - | 54.50 | 53.79 |
| ShapeLLM-13B [39] | arXiv,24.2 | 13B | 13.04B | 3D Point Cloud | - | - | 52.96 | - | - | 54.00 | 53.48 |
| **MiniGPT-3D** | - | 2.7B | **0.05B** (47.8M) | **3D Point Cloud** | **61.75** (+8.31) | **59.97** (+7.42) | **60.86** (+7.78) | **60.00** (+3.5) | **60.50** (+9.00) | **60.25** (+5.75) | **60.56** (+6.77) |

**Table 3: 3D object captioning results on Objaverse. The results are from human evaluation, GPT-4 evaluation, and traditional metrics. The bold and underline indicate the best and second best results, respectively.**

| Model | Reference | LLM Size | Trainable Params | GPT-4 | Sentence-BERT | SimCSE | Human Evaluation | | |
|---|---|---|---|---|---|---|---|---|---|
| | | | | | | | Correctness | Hallucination ↓ | Precision |
| InstructBLIP-7B [11] | NeurIPS,23 | 7B | 0.20B | 45.34 | 47.41 | 48.48 | 2.56 | 0.77 | 76.99 |
| InstructBLIP-13B [11] | NeurIPS,23 | 13B | 0.20B | 44.97 | 45.90 | 48.86 | 2.58 | 1.13 | 69.56 |
| LLaVA-7B [32] | NeurIPS,23 | 7B | 7.03B | 46.71 | 45.61 | 47.10 | 2.76 | 0.86 | 76.30 |
| LLaVA-13B [32] | NeurIPS,23 | 13B | 13.03B | 38.28 | 46.37 | 45.90 | 2.43 | 0.86 | 73.97 |
| 3D-LLM [20] | NeurIPS,23 | 13B | - | 33.42 | 44.48 | 43.68 | 1.77 | 1.16 | 60.39 |
| PointLLM-7B [51] | arXiv,23.8 | 7B | 7.01B | 44.85 | 47.47 | 48.55 | 3.04 | **0.66** | 82.14 |
| PointLLM-13B [51] | arXiv,23.8 | 13B | 13.01B | 48.15 | 47.91 | 49.12 | 3.10 | 0.84 | 78.75 |
| ShapeLLM-7B [39] | arXiv,24.2 | 7B | 7.04B | 46.92 | 48.20 | 49.23 | - | - | - |
| ShapeLLM-13B [39] | arXiv,24.2 | 13B | 13.04B | 48.94 | 48.52 | 49.98 | - | - | - |
| **MiniGPT-3D** | - | 2.7B | **0.05B** (47.8M) | **57.06** (+8.12) | **49.54** (+1.02) | **51.39** (+1.41) | **3.50** (+0.40) | 0.71 (+0.05) | **83.14** (+1.00) |

("gpt-3.5-turbo-0613") as our evaluation tools, like prior works [39, 51]. We choose multiple SOTA 3D-LLMs [18, 20, 39, 51] and two popular open-source 2D-LLMs [11, 32] as our baselines.

## 4.2 Generative 3D Object Classification

We conduct the generative 3D object classification tasks [51] on ModelNet40 [50] and Objaverse [12] datasets to assess MiniGPT-3D's categorical cognitive ability.

**Settings.** For a fair comparison, we utilize the classification evaluation settings similar to prior works [39, 51]. We employ identical prompts: the **I**nstruction-typed (I) prompt "What is this?" and the **C**ompletion-type (C) prompt "This is an object of". Point clouds and these prompts are fed into our MiniGPT-3D, outputting textual responses. For close-set zero-shot classification on ModelNet40, ChatGPT processes the text responses of MiniGPT-3D to select predicted categories from 40 ModelNet40 classes. For open-vocabulary classification on Objaverse, GPT-4 is employed as an evaluator to determine whether MiniGPT-3D's text response refers to the same category as the ground-truth caption.

**Results.** Experimental results are shown in Table 2. We achieve SOTA performance on all classification benchmarks using only one RTX 3090. Specifically, compared to the best baseline, ShapeLLM [39], we achieve significant improvements of 7.78% and 5.75% in average

accuracy on ModelNet40 and Objaverse datasets, respectively. Unlike other methods using LLM (7B or 13B) that require fine-tuning on 8 A100 or 8 A800 for hundreds of total GPU-hours, our MiniGPT-3D only utilizes a 2.7B LLM and trains 47.8M parameters on a single RTX 3090 GPU in 27 hours. These demonstrate the superiority and efficiency of our MiniGPT-3D, which leverages the powerful priors from 2D-LLMs to build 3D-LLM. Additionally, we observe that MiniGPT-3D exhibits the best performance in recognizing 3D objects of the ModelNet40 dataset unused during training, indicating its stronger generalization ability compared to other methods. Furthermore, even if using different prompts ( **I** and **C**) on the Objaverse dataset, MiniGPT-3D demonstrates highly consistent classification performance compared to other 3D-LLMs that have a 4% accuracy gap, showcasing its robustness for diverse prompts.

## 4.3 3D Object Captioning

To assess the model's understanding of 3D object details, we perform the 3D object captioning task.

**Settings.** For a fair comparison, we follow the evaluation settings of prior works [39, 51]. We use the prompt "Caption this 3D model in detail". We adopt three distinct evaluation methods: human evaluation, GPT-4 [35] evaluation, and traditional metric evaluation. In human evaluation, volunteers evaluate the model using standardized processes from PointLLM [51]. Specifically, focusing on object

**Table 4: Qualitative comparisons. The classification and caption results of models on ModelNet40 and Objaverse are presented. Our MiniGPT-3D generates more detailed and insightful responses compared to other baselines.**

| Samples 1, 2 | | | |
|---|---|---|---|
| Prompt | What is this? | | Briefly caption this 3D model. |
| Ground Truth | Laptop | | A cartoon black monster like a dragon |
| InstructBLIP-13B [11] | symbol letter l | | a black lizard with a sharp tooth in a dark room |
| LLaVA-13B[32] | A small, grainy, black and white letter j. | | A 3D model of a dark, menacing dragon. |
| 3D-LLM [20] | - | | A black and white tiger with long legs, standing on its hind leg. |
| Point-Bind LLM [18] | This is a laptop computer. | | The 3D model features a large, ornate gargoyle with a horned helmet, sitting on top of a building. |
| PointLLM-13B [51] | The 3D model represents a notebook computer, typically a laptop. | | The 3D model depicts a menacing black dragon, with its mouth opened wide revealing a row of sharp teeth. |
| **MiniGPT-3D** | **This is a 3D model of a sleek, black laptop. The laptop is rectangular in shape and has a glossy finish, suggesting a high-quality build. It has a flat screen, which is a common feature in modern laptops, and it's equipped with a keyboard and a touchpad for user interaction. The laptop appears to be in good condition, suggesting it could be used for various tasks such as work, entertainment, or education.** | | **This is a 3D model of a fearsome, black-colored dinosaur. The dinosaur possesses an aggressive stance, with sharp claws extending from its arms and legs. It has long black fangs that seem ready to attack.** |

**Table 5: Ablation on training process.**

| Row No. | Stage I | Stage II | Stage III | Stage IV | Acc. |
|---|---|---|---|---|---|
| 1 | ✓ | | | | 39.10 |
| 2 | ✓ | ✓ | | | 55.92 |
| 3 | ✓ | ✓ | ✓ | | 59.10 |
| 4 | ✓ | ✓ | ✓ | ✓ | **60.56** |
| 5 | ✓ | | ✓ | ✓ | 52.81 |
| 6 | | ✓ | ✓ | ✓ | 58.46 |
| 7 | ✓ | ✓ | | ✓ | 47.93 |

**Table 6: Ablation on 2D priors from 2D-LLM.**

| Modality Projector | Norm and LoRA for LLM | Acc. |
|---|---|---|
| | | 49.04 |
| ✓ | | 57.44 |
| | ✓ | 57.86 |
| ✓ | ✓ | **58.46** |

**Table 7: Ablation on stages using MQE.**

| Stage I | Stage II | Stage III | Stage IV | Acc. |
|---|---|---|---|---|
| ✓ | ✓ | ✓ | ✓ | 58.83 |
| | ✓ | ✓ | ✓ | 60.25 |
| | | ✓ | ✓ | 59.50 |
| | | | ✓ | **60.56** |

attributes (such as type, color, material, etc.), volunteers visually assess objects and assign correctness scores and hallucination scores to captions. Correctness measures model accuracy in describing attributes, while hallucination evaluates fabricated details' severity. Each attribute, correct or hallucinated, receives a point. Precision is calculated as the ratio of correct information in model-generated content. The Inter-Annotator Agreement score is 0.89 on ICC1k, indicating volunteers' high consistency in cognitive understanding and scoring criteria. GPT-4 evaluates semantic similarity between our model's output and manually annotated captions. In traditional metric evaluation, like prior works [39, 51], we use data-driven metrics like Sentence-BERT [41] and SimCSE [17], instead of BLEU-1 [37], ROUGEL [28], and METEOR [3], because the latter lack sufficient capabilities in semantic evaluation.

**Results.** As shown in Table 3, our MiniGPT-3D achieves SOTA performance on multiple metrics. Specifically, MiniGPT-3D outperforms ShapeLLM-13B [39], by a large margin of 8.12 on the GPT-4 evaluation score, setting new SOTA with only 2.7B LLM, indicating robust 3D detail comprehension. Also, compared to ShapeLLM-13B, MiniGPT-3D surpasses 1.02 and 1.41 on Sentence-BERT and SimCSE metrics, respectively, achieving new SOTA with its remarkable ability to generate accurate captions matching ground truth. Human evaluation further reveals MiniGPT-3D's superior correctness and precision scores compared to baselines. Notably, even with a 2.7B LLM, MiniGPT-3D exhibits a hallucination score comparable to SOTA, surpassing larger 13B LLM-based methods. These outstanding results showcase MiniGPT-3D's fine-grained understanding of 3D objects, inheriting the cognitive capabilities of 2D-LLM.

## 4.4 Qualitative Results

Figure 1(e) qualitatively shows the MiniGPT-3D's powerful ability to perceive 3D object details. Our MiniGPT-3D precisely extracts information from 3D objects, encompassing categories, colors, shapes, materials, and internal component relationships. Additionally, MiniGPT-3D can perform reasonable reasoning based on object cues, such as potential occurrence periods and locations. Figure 1(f) further demonstrates MiniGPT-3D's comprehension of 3D object information in open-ended dialogues. MiniGPT-3D accurately outputs 3D object-related world knowledge, showcasing its extensive textual knowledge inherited from LLMs.

In sample 1 of Figure 4, our MiniGPT-3D successfully recognizes the shape, screen, and keyboard of a laptop, compared to other methods. Furthermore, it can deduce the potential usage of this 3D object. In the more complex sample 2 of Figure 4, our MiniGPT-3D demonstrates superior understanding capabilities of 3D objects by recognizing additional features like the dinosaur's sharp claws and inferring its potential action intentions, compared to other methods.

## 4.5 Ablation Studies

In this section, we conduct ablation studies to investigate various model design options. Herein, we report the total average accuracy of MiniGPT-3D on the generative classification benchmark.

**4.5.1 Training process.** We conduct ablation study to validate the efficacy of our four-stage training strategy. The results in Table 5 highlight the optimal performance achieved by our approach. Specifically, comparing Row #4 vs. #6, we observe that the first

**Table 8: Ablation on fine-tuned modules in Q-Former.**

| LoRA Q, K, V | LoRA Dense | Norm | Acc. |
|:---:|:---:|:---:|:---:|
| | | | 58.18 |
| ✓ | | | 59.85 |
| ✓ | ✓ | | 59.97 |
| ✓ | ✓ | ✓ | 60.14 |
| ✓ | | ✓ | **60.56** |

**Table 9: Ablation on the number of query experts.**

| Number | Acc. |
|:---:|:---:|
| 1 | 59.19 |
| 3 | 59.66 |
| 6 | 59.14 |
| 8 | **60.56** |
| 10 | 59.85 |

**Table 10: Ablation on the point cloud projection layers.**

| Number of layers | Acc. |
|:---:|:---:|
| 1 | 57.02 |
| 2 | **60.56** |
| 3 | 59.20 |

**Table 11: Ablation on router type of MQE.**

| Type | Acc. |
|:---:|:---:|
| Constant Router | 60.10 |
| Sparse Router | **60.56** |
| Soft Router | 60.31 |

**Table 12: Ablation on trained modules in stage IV.**

| MQE | Norm. & LoRA for Q-Former | Modality Projector | Norm. & LoRA for LLM | MLP | Acc. |
|:---:|:---:|:---:|:---:|:---:|:---:|
| ✓ | ✓ | ✓ | ✓ | ✓ | 58.93 |
| ✓ | ✓ | ✓ | ✓ | | 59.93 |
| ✓ | ✓ | ✓ | | | 59.02 |
| ✓ | ✓ | | | | 59.64 |
| ✓ | | | | | **60.56** |

stage bridges knowledge between 2D-LLM and 3D encoder, enabling smoother semantic transitions across different dimensional spaces. Comparing Row #4 vs. #5, we note that the second training stage which involves using easy tasks to adapt the knowledge of the 2D-LLM to the 3D space, allows the model to focus on enhancing cognitive capabilities in subsequent stages. Comparing Row #4 vs. #7, the third training stage utilizes more challenging tasks to reinforce the newborn 3D cognitive abilities, providing a reliable semantic context for the final stage to train MQE. Comparing Row #4 vs. #3, the inclusion of the fourth stage, dedicated to training the MQE, enables each query expert to acquire unique knowledge, further enhancing MiniGPT-3D's understanding of 3D objects.

*4.5.2* ***2D priors from 2D-LLM****.* We conduct ablation study to varify the effectiveness of the 2D priors from 2D-LLM, as detailed in Table 6. Since dropping any pre-trained weights of 2D-LLM would make the first training stage infeasible, all cases of this ablation study are just trained through stages II to IV. We find that removing any of 2D-LLM weight degrades performance, and discarding more pre-trained weights of 2D-LLM causes an up to 9.4% accuracy drop. These results highlight the crucial role of 2D-LLM knowledge in boosting 3D-LLM performance. Using 2D-LLM modules facilitates cost-efficient training of 3D-LLM even on consumer GPUs like RTX 3090 GPU, enhancing accessibility for the community.

*4.5.3* ***Training stages using MQE****.* We further investigate the impact of training MQE in different stages, with detailed results presented in Table 7. Our results indicate that introducing MQE in only stage IV achieves optimal performance. The I-III stages enable the model to learn enough semantic features, paving the way for MQE to adaptively select useful information in stage IV.

*4.5.4* ***Fine-tuned modules in Q-Former****.* Employing PEFT methods to fine-tune Q-Former can better align point features with LLM, avoiding expensive computation. As outlined in Table 8, fine-tuning the Query, Key, and Value layers with LoRA [21], along with normalization layers, maximizes the potential of Q-Former. Notably, we efficiently fine-tune the 105M-parameter Q-Former using only 0.7M parameters, achieving a 2.38% accuracy improvement compared to the frozen Q-Former.

*4.5.5* ***Number of query experts****.* Within MQE, each query expert holds unique knowledge, facilitating extraction of point cloud features. Our experiments, in Table 9, reveal that 8 query experts yield optimal performance. Insufficient experts may compromise information extraction, while excessive ones may affect cooperation among experts. Notably, single-expert, i.e. without MQE, results in a 1.37% accuracy drop, highlighting the superiority of MQE.

*4.5.6* ***Point cloud projection layer****.* The point cloud projection layer bridges point cloud features with the 2D semantics of frozen Q-Former, while ensuring dimensional alignment. As shown in Table 10, our experiments demonstrate that two MLP layers offer the optimal setup, as excessive or insufficient layers can result in information loss, compromising overall performance.

*4.5.7* ***Router type of MQE****.* The routing mechanism in MQE regulates the cooperation among query experts. The constant router [25] assigns static average weights, while the soft router [38] dynamically assigns weights during training. The sparse router [43] selects the top two experts based on the dynamic weights provided by the soft router. We explore these router types in Table 11, finding that the sparse router, which dynamically assigns weights and selects the most promising experts, maximizes the capabilities of MQE.

*4.5.8* ***Trained modules in stage IV****.* In the training stage IV, only MQE is trained to enable each query expert to learn knowledge within a stable semantic context. Our experiments in Table 12 investigate the integration of various training modules. The results indicate that stage IV is to adaptively aggregate features of different experts, with knowledge gained from I-III stages frozen. Losing any frozen knowledge causes information loss, demonstrating the MQE is specifically designed for information aggregation.

# 5 CONCLUSION

In this paper, we present MiniGPT-3D, a efficient and powerful 3D-LLM, requiring the training of only 47.8M learnable parameters within 26.8 hours on one single NVIDIA RTX 3090 GPU. Specifically, we propose a novel four-stage training strategy that gradually aligns 3D point cloud features with LLM using 2D priors from 2D-LLM. Additionally, we design the mixture of query experts, introducing MoE to queries, to adaptively aggregate multiple features. Extensive experiments show the superiority of MiniGPT-3D in 3D point cloud understanding and question answering.

**Discussion.** MiniGPT-3D's limitations lie in its training on object-level datasets, preventing it from understanding large-scale point clouds. Moreover, like existing 3D-LLMs, our MiniGPT-3D solely focuses on comprehending static 3D objects, lacking the capacity to recognize the actions of dynamic objects. We will extend our 3D-LLM building approach to autonomous driving scenarios.

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
