# OpenReview forum: "MiniGPT-3D: Efficiently Aligning 3D Point Clouds with Large Language Models using 2D Priors"
_acmmm.org/ACMMM/2024/Conference — MM2024 Poster_

### Official Review · Reviewer_ns8X · 2024-05-19

**Rating:** 6
**Confidence:** 4

**Summary:**

This paper addresses the challenge of integrating 3D point clouds into Large Language Models (LLMs) in a cost-effective manner. While Large 2D vision-language models (2D-LLMs) have demonstrated success in combining text and images, aligning 3D data with LLMs incurs substantial training costs. The authors propose MiniGPT-3D, a novel approach that achieves state-of-the-art (SOTA) results while significantly reducing training time and computational resources.

MiniGPT-3D leverages 2D priors from 2D-LLMs to align 3D point clouds with language models efficiently. The paper introduces a four-stage training strategy for modality alignment and a mixture of query experts module to aggregate features adaptively. Additionally, parameter-efficient fine-tuning methods such as LoRA and Norm fine-tuning are employed, resulting in a model with significantly fewer learnable parameters compared to existing approaches.

Experimental results demonstrate that MiniGPT-3D outperforms previous methods on 3D object classification and captioning tasks, achieving superior performance at a fraction of the training cost.

**Strengths:**

1. Novelty in Approach: The paper introduces MiniGPT-3D, a novel approach for efficiently aligning 3D point clouds with Large Language Models (LLMs) by leveraging 2D priors from 2D-LLMs. This novel approach significantly reduces the training costs associated with integrating 3D data into language models, addressing a critical challenge in the field.

2. Efficiency and Cost-Effectiveness: MiniGPT-3D achieves state-of-the-art results while requiring substantially fewer computational resources compared to existing methods. By training for only 27 hours on a single RTX 3090 and employing parameter-efficient fine-tuning methods, the model demonstrates high efficiency and cost-effectiveness, making it accessible for researchers with limited computational resources.

3. Comprehensive Evaluation: The paper conducts extensive experiments to evaluate MiniGPT-3D's performance on 3D object classification and captioning tasks. The results demonstrate significant improvements over existing methods, showcasing the effectiveness of the proposed approach across different applications.

**Limitations:**

Limited Scope to 3D Objects: While the proposed MiniGPT-3D model demonstrates impressive performance in aligning 3D point clouds with language models, its applicability is restricted to 3D object-related tasks. The paper does not address the broader challenge of integrating 3D scenes or complex environments into language models.
However, it's important to note that this limitation is not unique to the proposed approach but rather a common challenge faced by current 3D Multi-Modal Language Models (MLLMs).

**Suitability:**

3

---

### Official Review · Reviewer_dVEG · 2024-05-20

**Rating:** 4
**Confidence:** 3

**Summary:**

The paper presents MiniGPT-3D, an efficient 3D point cloud-language model (3D-LLM) that leverages 2D vision-language models (2D-LLMs) as priors to align 3D point clouds with LLMs. The authors propose a four-stage training strategy that progressively transfers knowledge from 2D-LLMs to the 3D space, requiring only 27 hours of training on a single RTX 3090 GPU. MiniGPT-3D introduces a Mixture of Query Experts (MQE) module for feature aggregation and uses parameter-efficient fine-tuning methods, resulting in a model with significantly fewer trainable parameters than existing methods. The model achieves state-of-the-art (SOTA) results on 3D object classification and captioning tasks with substantially reduced training costs.

**Strengths:**

Efficiency: MiniGPT-3D demonstrates a highly efficient training process, requiring minimal GPU hours and leveraging a consumer-grade GPU (RTX 3090), making it accessible for more researchers.
Performance: The model achieves SOTA results on generative 3D object classification and captioning tasks, outperforming other methods with less computational overhead.

**Limitations:**

1. Integrating 2D modality to connect point cloud and text modalities isn't a novel approach. ULIP[1]/CG3D[2]/OpenScene[3] has utilized similar methodologies to synchronize point cloud attributes with the features of the CLIP foundation model. A thorough comparative analysis would be beneficial to understand the advancements or improvements over previous methods.

2. The absence of testing on more intricate datasets, such as ScanNet, is noticeable. OCTAVIUS[4] has endeavored to develop a multimodal Large Language Model (LLM) to interpret complex scenes and perform Captioning/Visual Question Answering (VQA) tasks on the ScanNet dataset. This suggests a need for experimentation on such complex datasets to validate the robustness and versatility of the model.

3. The introduction of LoRA-MoE by Octavius marks an interesting development. Discussing the relationship between MQE and LoRA-MoE requires a multifaceted approach, considering aspects like model efficiency, knowledge retention, and adaptability to various tasks.

4. The four-stage training process appears intricate, and providing more detailed information about each stage, such as the duration of training and specific processes involved, would be essential for a comprehensive understanding of the model's learning efficiency.

[1] Xue L, Gao M, Xing C, et al. Ulip: Learning a unified representation of language, images, and point clouds for 3d understanding[C]//Proceedings of the IEEE/CVF Conference on Computer Vision and Pattern Recognition. 2023: 1179-1189.

[2] Hegde D, Valanarasu J M J, Patel V. Clip goes 3d: Leveraging prompt tuning for language grounded 3d recognition[C]//Proceedings of the IEEE/CVF International Conference on Computer Vision. 2023: 2028-2038.

[3] Peng S, Genova K, Jiang C, et al. Openscene: 3d scene understanding with open vocabularies[C]//Proceedings of the IEEE/CVF Conference on Computer Vision and Pattern Recognition. 2023: 815-824.

[4] Chen Z, Wang Z, Wang Z, et al. Octavius: Mitigating task interference in mllms via moe[C] ICLR2024

**Suitability:**

3

---

### Official Review · Reviewer_ZEW2 · 2024-05-24

**Rating:** 4
**Confidence:** 2

**Summary:**

The paper introduces MiniGPT-3D, an innovative 3D point cloud-language model (3D-LLM) that leverages pre-trained 2D vision-language models (2D-LLMs) as a prior for more efficient training. Unlike traditional 3D-LLMs that require extensive resources, MiniGPT-3D achieves state-of-the-art (SOTA) results on 3D object classification and captioning tasks with significantly reduced resource usage and training time. The model uses a novel four-stage training strategy and a Mixture of Query Experts (MQE) module to adaptively aggregate features, employing parameter-efficient fine-tuning methods such as LoRA and Norm fine-tuning.

**Strengths:**

1. Efficiency in Training: MiniGPT-3D drastically reduces the training resources and time required compared with previous models like ShapeLLM13B.
2. Well-designed Training Strategy: The paper introduces a cascaded four-stage training strategy that effectively aligns 3D point cloud data with LLMs using 2D priors. This method facilitates a more efficient transfer and integration of visual-textual representations from 2D to 3D.

**Limitations:**

1. The 2D priors are embedded within 2D-LLM parameters without a clear method for aligning 2D and 3D representations, such as using multi-view images of point clouds. This lack of detailed alignment strategy brings into question how well the 3D model utilizes the 2D information.

**Suitability:**

2

---

### Official Review · Reviewer_MiBh · 2024-05-25

**Rating:** 4
**Confidence:** 3

**Summary:**

The architecture consists of several components: a 3D encoder, a 3D projection layer, a Query Transformer (Q-Former), a mixture of query experts, a modality projector, and a large language model (LLM). The training process is strategically divided into four stages to progressively align 3D embeddings with language embeddings. Despite its minimal number of trainable parameters, this model achieves efficient training and superior performance compared to existing methods in 3D object classification and captioning.

**Strengths:**

1. The paper presents a complex technical design featuring multiple modules and sequential training stages, illustrating a sophisticated approach to aligning 3D point clouds with language.
2. The experiment and ablation sections are comprehensive. Discussions and comparisons with related works are thorough.
3. The paper is well-written and easy to follow.

**Limitations:**

1. The modules introduced in this paper, such as the modality projector, Mixture of Experts (MoE), and Q-Former, are well-established in the field of VLMs. Only reusing these familiar techniques may lead to perceptions of limited novelty at the idea level within this paper.
2. I am curious about the network’s capabilities in open-world settings or its understanding at the scene level. Does the proposed model demonstrate any advancements in these aspects?
3. In the object captioning experiments, the MiniGPT-3D achieves a notably higher GPT-4 score compared to other methods, yet the improvement in sentence semantic metrics (such as Sentence-BERT and SimCSE) is marginal. Could you provide an analysis or discussion on this phenomenon? What might explain the significant disparity in performance across these metrics?

**Suitability:**

3

---

### Meta-Review · Area_Chair_Zjzk · 2024-06-28

**Recommendation:** Accept (Poster)
**Confidence:** 5

**Metareview:**

The paper introduces a 3D-language model that leverages pre-trained 2D vision-language models (2D-LLMs) as a prior for more efficient training. All of the reviewers agree that it is a nice paper. The AC also agrees with their comments and recommends the paper be accepted.